# Study of the Effect of L-PBF Technique Temporal Evolution on Microstructure, Surface Texture, and Fatigue Performance of Ti gr. 23 Alloy

Alex Lanzutti *, Michele Magnan ⬤, Emanuele Vaglio ⬤, Giovanni Totis ⬤, Marco Sortino ⬤ and Lorenzo Fedrizzi

Polytechnic Department of Engineering and Architecture, University of Udine, Via delle Scienze 208, 33100 Udine, Italy
* Correspondence: alex.lanzutti@uniud.it; Tel.: +39-0432-629978

**Abstract:** Titanium alloys are widely used in various technological fields due to their excellent performance. Since the early stages of the 3D printing concept, these alloys have been intensively used as materials for these processes. In this work, the evolution of the performance of the 3D printing process has been studied by analysing the microstructure and the mechanical properties, fatigue and tensile, of the Ti gr. 23 alloy produced by two different models of Concept Laser M2 Cusing machines (an old model and a more recent one). The process parameters recommended by the manufacturer were adopted for each machine. Both microstructural and surface texture characterisations were carried out to better correlate the differences with the production process technique. For the same purpose, tensile tests and microhardness profiles were obtained, while the dynamic mechanical properties were evaluated by means of fatigue tests aimed at determining the fatigue limit of the material using a staircase approach. The mechanical tests were carried out on specimens with three different orientations with respect to the building platform, using two different SLM techniques. The fatigue behaviour was then analysed by evaluating the fracture surfaces and, in particular, the crack nucleation sites. By comparing the calculated fatigue values with the results of local fatigue calculations, an estimate of the residual stresses near the crack nucleation site was obtained. The results showed that the specimens produced on a newer machine had lower roughness (about 10%), slightly higher ductility, and a higher fatigue limit (10–20 MPa) compared to the specimens produced with the same material but on older equipment.

**Keywords:** L-PBF; fatigue resistance; Ti alloy; surface texture; fatigue limit

## 1. Introduction

In the last few years, additive manufacturing techniques for metal components have evolved rapidly in terms of applied technology, with the aim of continuously improving the surface quality, the microstructure, and the mechanical properties of the 3D printed parts.

Titanium alloys are widely studied materials because of their excellent corrosion resistance and good mechanical properties. Nevertheless, the cost of these alloys is high, and their production by means of conventional subtractive machining techniques is expensive because of the material losses, tool consumption, demanding CAM/CNC programming phase, and relatively high machining time due to their low machinability. The additive manufacturing techniques are very promising since they may allow significant savings in terms of lead time and costs for these specific alloys.

Among titanium alloys, Ti gr. 5 is commonly adopted in many strategic fields such as aerospace and energy, while Ti gr. 23 is mainly applied in the biomedical field. The main difference between these alloys is that Ti gr. 23 has a low amount of interstitial elements with respect to Ti gr. 5. Both of them are processed by several additive manufacturing techniques such as L-PBF (Laser-Powder Bed Fusion) [1–32] and EBM (Electron Beam

Melting) [8,15,33–35]. Many research works study the effects of process parameters on internal defect generation [22,33,34,36], surface texture [13,30,31], and microstructure [21,34]. Accordingly, process parameters have a strong influence on the final mechanical performances [3,7,10,21,30,31,33,36]. In detail, Khorasani et al. [10] showed that an increase in power density produced a good material microstructure with high hardness, while increasing the hatch spacing reduced the hardness. A more detailed study on the effect of process parameters on microstructural texture was performed by Zhao et al. [21]. They observed that low laser power and low scan rates produced coarser columns of Ti martensite. On the contrary, an increase in both parameters produced a less textured martensitic microstructure, thus also influencing the final mechanical properties of the material. Some studies were also devoted to determining the effect of surface texture on mechanical properties [10]. A study more dedicated to the effect of defects on microstructure and mechanical properties was performed by Gong et al. [33]. In this case, they observed that the amount and dimension of internal voids in 3D printed samples have no influence on the microstructure of the material and are strongly dependent on the process parameters. It was recently observed that samples' orientation with respect to the building platform has a strong impact on the mechanical properties as well as on microstructure and surface texture [2,4–6,9,24,28,32]. Concerning the tensile tests, both vertical and 45° printing directions show slightly higher mechanical properties compared to the horizontal samples in terms of Tensile Strength (TS), Yield Strength (YS), and Elongation (E%) [5,16,24,32].

Going into the details of the research, Sun et al. [24] observed that the highest tensile strength was reached by samples oriented at 45°. The materials were heat-treated before the tests. On the other hand, a strong influence of defects on mechanical properties was observed. The effect of process parameters on tensile tests is mainly related to grain orientation, microstructure, and defect density. Barba et al. observed that the mechanical properties are strongly influenced by the macro-columnar grain's orientation, which depends on growth direction and is also responsible for certain anisotropy in the material. They also observed a dependence of the mechanical response of the material on alpha lath orientation. Simonelli et al. [32], instead, stated that there is an influence on the mechanical response of material due to prior beta boundary orientation and distribution.

Many studies focused on the fatigue performances of the 3D printed samples produced with Ti gr. 5 or Ti. gr. 23 alloys [1,6,8,12,13,37,38]. Fatigue performance was investigated by varying the samples' orientation, surface finishing techniques, and post printing heat treatments. In most cases, the 3D printed samples were machined before the fatigue life tests. Only a few researchers measured the samples in their as-built condition. Low Cycle Fatigue (LCF tests) [1], High Cycle Fatigue (HCF) [1,12,19,39] or fatigue crack propagation [4,27] were examined. As a result, fatigue life strongly depends on the number of defects and on the orientation of samples with respect to the building platform. In particular, the main defects responsible for fatigue failure of the specimens are close to the external surface in samples with a printed surface, as observed by Benedetti et al. [1]. In machined samples, the defects responsible for material failure are internal and usually have a strong influence on the number of cycles to failure. In detail, the bigger the defect size, the lower the material resistance in terms of the number of cycles. Surface roughness may also play a role in the fatigue properties of 3D printed materials [13]. In particular, the printing process may produce some surface dross in proximity to the valley generated by layer-by-layer production, which is also responsible for the increase in roughness in samples produced in the vertical direction [6]. However, a major role in fatigue life is played by residual stresses [1,4,20]. In particular, the residual stresses in printed samples are in a tensile condition, which can reduce the fatigue life of the material, as observed by Benedetti et al. [1] and Syed et al. [4]. This was also assessed by studying the effects of the stress-relieving treatment on fatigue life [14]. However, the stress-relieving heat treatment may alter the material microstructure [40], thus further influencing the final result. Nevertheless, no studies on both the effect of printing machine evolution and samples' orientation on the fatigue properties of as-built specimens (not stress-relieved) were found in the literature.

Considering the scarce literature on the effect of temporal evolution of L-PBF 3D printing machines, this work aims to analyse and then correlate the effect of machine temporal evolution on both microstructure, surface texture, and mechanical properties. In order to achieve this goal, a detailed microstructural characterisation will be carried out in order to determine both the effect of the temporal evolution on the internal defect distribution and then on the microstructure. For this purpose, a precise evaluation is carried out by means of microscopic techniques.

The surface texture is then studied by studying the surface roughness observed on samples produced with different printing directions and L-PBF techniques.

The effect of machine obsolescence was also evaluated by means of tensile tests and fatigue tests by evaluating YS, TS, E% (for tensile tests), and fatigue limit (for fatigue tests). An attempt is made to correlate the fatigue properties with both surface texture and microstructure using empirical models.

Measurements have been carried out on as-built samples without any post-print stress-relieving thermal treatment, which may be responsible for altering the microstructure of the material, also taking into account the effect of different printing directions on the above properties.

## 2. Materials and Methods

### 2.1. Sample Preparation

A commercial Ti gr. 23 powder was used for producing 3D printed samples. The chemical composition, acquired by optical spectroscopy and gas in metal analysis, is given in Table 1.

**Table 1.** Chemical composition (wt%) of the Ti powders used in this experiment.

| | Al | V | Fe | C [ppm] | O [ppm] | N [ppm] | H [ppm] | Ti |
|---|---|---|---|---|---|---|---|---|
| Powders | 5.7 | 3.8 | 0.22 | 50 | 76 | 32 | 5 | Bal. |
| Chemical composition range | 5.5–6.5 | 3.5–4.5 | Max 0.25 | Max 80 | Max 130 | Max 50 | Max 12 | Bal. |

The Ti gr. 23 powder consists of spherical particles (Figure 1) with a granulometric distribution ranging from 8.3 μm (10th %ile) to 40.0 μm (90th %ile) and a median diameter of 23.3 μm (Figure 1a). No defects such as voids or blows could be detected in the metal powders (embedded in epoxy resin and metallographically prepared), which present a Ti martensite microstructure.

The samples were manufactured using 2 different Concept Laser M2 Cusing machines, one installed in 2013 (older) and the other in 2017 (newer). Both machines were equipped with a single-mode CW ytterbium-doped fibre laser with an emission wavelength of 1070 nm and processed the samples under an inert argon atmosphere with less than 0.2% residual oxygen. The main difference between the machines was the optical system, which in the older model was composed of the collimator, the beam expander, the scanning head, and the F-theta lens, while in the newer model it also included a dynamic focusing unit for adjusting the laser spot size. Consequently, radically different process parameters (see Table 2) were recommended by the machine manufacturer when processing the considered titanium alloy in the two machine types. In particular, the optical system of the older machine forced the use of a small laser beam in a defocused condition, while the focusing unit installed in the newer machine enabled the use of a much larger laser beam in a focused condition. Consequently, the other recommended process parameters differ from each other both when compared one at a time and when combined in synthetic indicators describing the energy density. The most similar physical quantity in the two cases was the Linear Energy Density (LED) [22], which was 17% higher on the bulk areas and 27% lower on the contours with the new machine. The Surface Energy Density (SED) [35] and the Volumetric Energy Density (VED) [23] were instead remarkably lower. However, the energy density

is not a good indicator for comparing different process parameters [41], since it neglects the main effects and interactions among the considered factors. In addition, it also fails to capture the influence of defocusing, which is a main feature of the tested equipment. For those reasons, it was decided to investigate the effect of technological advances in LPBF on the properties of Ti gr. 23 components by comparing the performance of the machines in their respective optimum conditions instead of equalling the energy input.

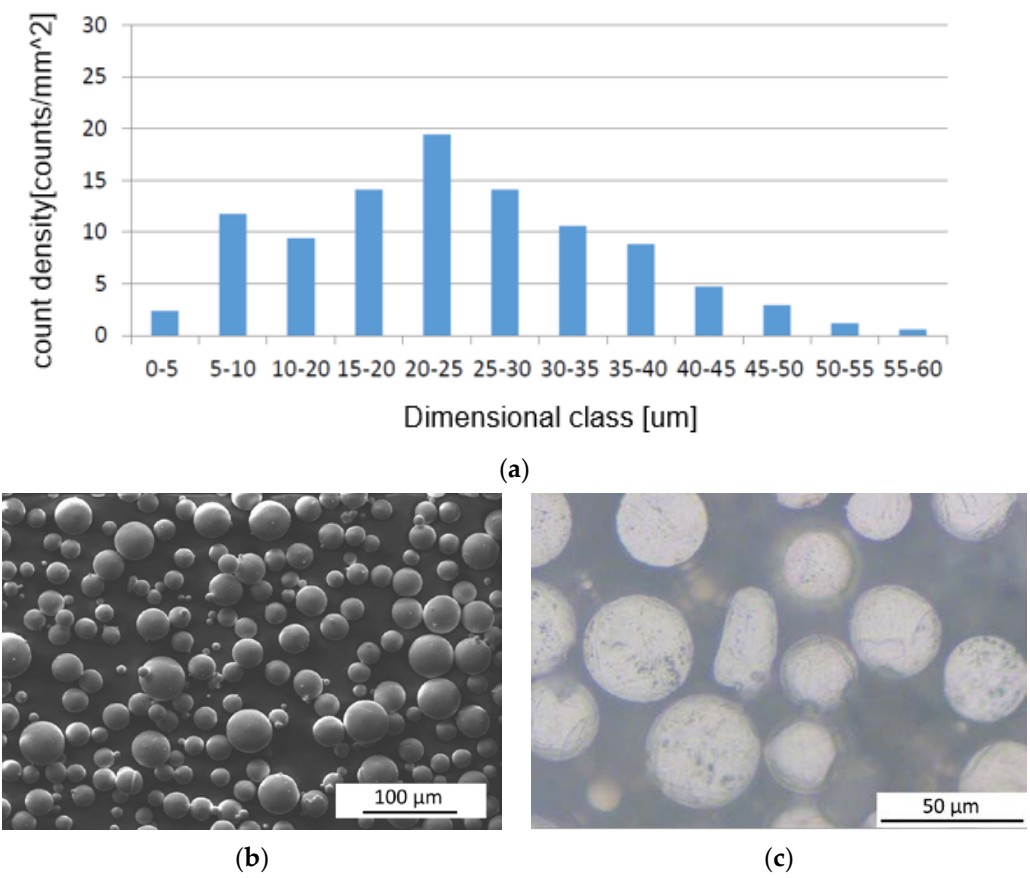

**Figure 1.** (**a**) Progressive granulometric distribution of the Ti gr. 23 alloy powder, measured by SEM. (**b**) SEM image of the powders (magnification = 500×). (**c**) Light microscope analysis of the powders.

**Table 2.** Process parameters used for the samples manufacture.

|  | Older Equipment | | Newer Equipment | |
|---|---|---|---|---|
|  | Specimens | Contour | Specimens | Contour |
| Power | 180 W | 180 W | 225 W | 200 W |
| Scanning speed | 1250 mm/s | 1250 mm/s | 1300 mm/s | 1750 mm/s |
| Spot diameter | 50 μm | 50 μm | 155 μm | 70 μm |
| Hatch distance | 105 μm | - | 90 μm | - |
| Layer thickness | 30 μm | 30 μm | 25 μm | 25 μm |
| Defocusing distance | −1.8 mm | −1.8 mm | 0 mm | 0 mm |
| LED | 0.14 J/mm | 0.14 J/mm | 0.17 J/mm | 0.11 J/mm |
| SED | 2.88 J/mm$^2$ | 2.88 J/mm$^2$ | 1.12 J/mm$^2$ | 1.63 J/mm$^2$ |
| VED | 96.00 J/mm$^3$ | 96.00 J/mm$^3$ | 44.66 J/mm$^3$ | 65.31 J/mm$^3$ |

Similarly, the samples were scanned according to the specific strategy recommended by the machine manufacturer for each machine model. In more detail, the samples produced with both machines were scanned according to the island exposure strategy, which consists of a chessboard pattern of 5 × 5 mm squares that were bi-directionally scanned along mutually perpendicular directions (Figure 2). Adjacent squares were spaced by 105 μm on the older machine and by 96 μm on the newer machine, while in both cases they underwent an angular shift of 90° and an XY shift of 1 mm at each layer in order to guarantee the material uniformity.

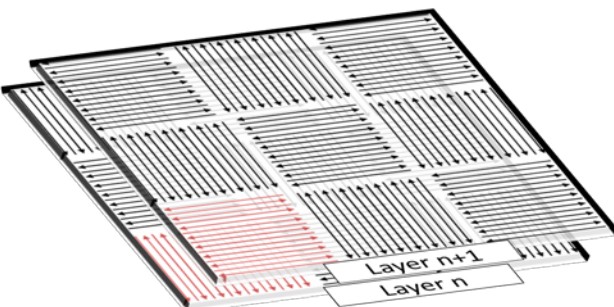

**Figure 2.** Island exposure strategy used for producing the experimental samples.

Finally, in both cases, the contouring operation consisted of exposing a single curve along the perimeter of the parts sections. The contour was shifted inward by 0.075 mm from its nominal position and was aimed at improving the surface quality.

A total of 60 dog-bone (Figure 3) samples were prepared by each machine in multiple build jobs. One third of them were oriented parallel to the building platform (0°), one third were oriented perpendicular to it (90°), and one third were built with an intermediate orientation (45°). The specimens were randomly arranged on the building platform in order to avoid any possible dependence on their spatial position.

No stress relieving heat treatment was carried out after the printing process. The fatigue samples were directly detached from the building platform and machined in proximity to the gripping head in order to avoid misalignments during the fatigue tests. The supported surfaces of the samples oriented at both 0° and 45° were manually ground with abrasive papers in order to avoid the surface texture effect caused by the presence of support residuals.

### 2.2. Microstructural Characterisation and Microhardness Measurements

The dog-bone specimens that were realised for mechanical tests were also used for microstructural characterisation. The samples for microstructural characterisation were derived from the dog-bone cross section in the gauge length. They underwent hot mounting in epoxy resin followed by a standard metallographic preparation in order to obtain a mirror-like surface (last polishing step with colloidal silica). The internal voids were characterised and classified on the whole area of the samples by means of light microscope analysis at a fixed magnification (100×). Afterwards, a distribution of the defect density as a function of their dimensions was performed.

The samples, after the void classification, were etched using Kroll etchant (30 s) and then analysed by a light microscope near the external surface and the bulk.

On the same specimens, a microhardness profile was acquired across the radius (HV 0.3 with a distance between each measurement of 0.5 mm).

### 2.3. Surface Morphology and Roughness

The roughness (Ra, Ry, Rz, and ρr (average curvature of the valley of roughness calculated in agreement with Arola et al. [42])) values of the samples were measured using a stylus contact profilometer (Dektak-150) with a stylus tip of 12 μm radius. The $K_t$

related to surface topography was calculated by means of the formula suggested by many authors [43], that is:

$$K_t = 1 + 2 \left( \frac{R_a}{\overline{\rho_r}} \right) \left( \frac{R_y}{R_z} \right) \tag{1}$$

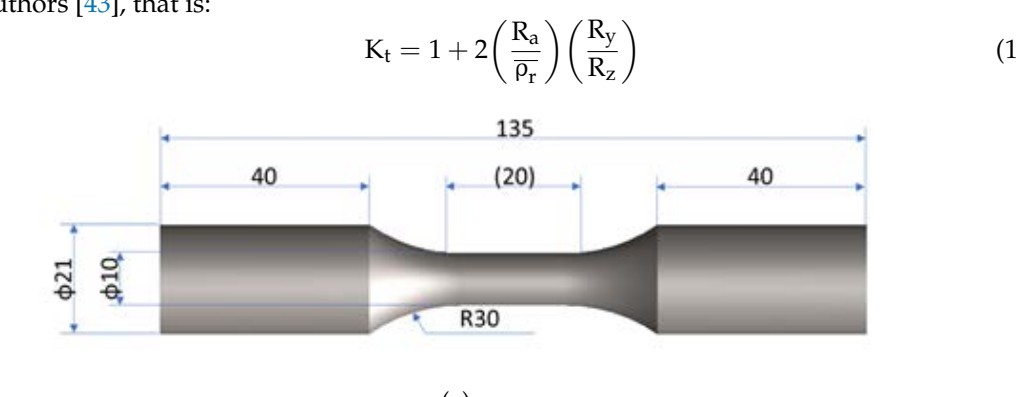

(**a**)

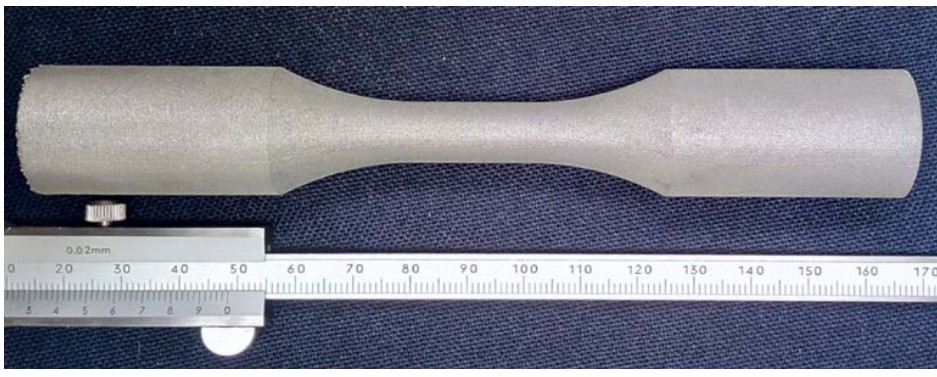

(**b**)

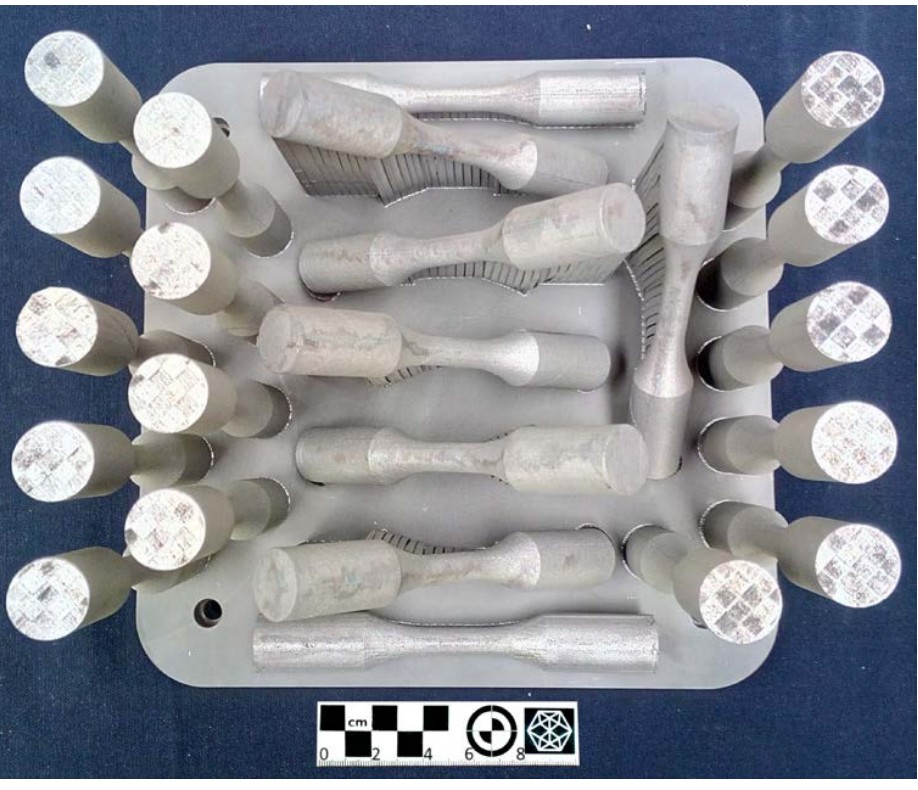

(**c**)

**Figure 3.** Drawing of the specimens printed in this experiment (**a**), example of a 3D printed specimen (**b**), and representative job with the printed samples (**c**).

*2.4. Tensile and Fatigue Tests*

The mechanical properties of the produced samples were determined by means of both tensile and fatigue tests at room temperature. The tensile tests were performed under displacement control by applying a displacement rate of 1 mm/min. After the rupture of the specimen, the Tensile Strength (TS), Yield Strength (YS), and Elongation (E%) were acquired. The tests were performed on 3 specimens for each printing direction and for each 3D printing technique used in this work.

Regarding the fatigue tests, the staircase approach was used to determine both the fatigue data dispersion and the fatigue life endurance. The tests started at a stress level of 200 MPa, close to the fatigue limit found by other researchers [1,14], and the stress variation ($\Delta$S) was $\pm$ 25 MPa (in the case of failure, the stress variation was decreased, while in the case of runout, the stress was increased). The runout cycles were $10^6$. Concerning the fatigue test results, S-N diagrams were obtained and analysed for each printing direction and instrument used to produce the sample. The fatigue limit was calculated using the Dixon–Mood approach [43]. Afterwards, the failed specimens underwent SEM analysis of the fracture surfaces in order to determine the crack nucleation site. In this case, by combining the microhardness results with the extension of the fatigue crack, the local fatigue limit ($\sigma_{w0}$) in proximity to the discontinuity that generated the fatigue crack was calculated according to the Murakami approach [44]. The formula used for surface crack nucleation sites was:

$$\sigma_{w0} = 1.43 \cdot \left( \frac{HV + 120}{\left( \sqrt{area} \right)^{1/6}} \right) \tag{2}$$

The local fatigue limit was then compared to the fatigue limit ($\sigma_{lf}$) found by the Dixon–Mood approach in order to have rough information on the residual stress level that affected the fatigue performance of the material. In particular, as suggested by other authors [2], Walker's model was used to estimate the $R_{real}$ (ratio between $\sigma_{min}$ and $\sigma_{max}$ considering also the residual stresses), whereas the model was:

$$\frac{\sigma_{lf}}{\sigma_{w0}} = \left( \frac{1 - R_{real}}{2} \right)^n \tag{3}$$

The n coefficient for the as-built samples in Ti alloy was equal to 0.385. This equation was used to qualitatively evaluate the residual stresses in proximity to the crack nucleation site.

## 3. Results and Discussion

*3.1. Microstructural Characterisation and Microhardness Measurements*

The microstructural characterisation of the samples starts with the analysis of the pore and void distribution found on specimens printed by either machine (Figure 4). In this case, the data presented are a summary of the results found for each printing direction. However, the effect of the printing machines is also clearly visible.

The void/pore density is higher in the specimens produced by the older equipment compared to the specimens produced by the newer equipment for each dimensional class. All the specimens present a prevalence of defects smaller than 40 µm. The observed difference among the results obtained with the two machines is mainly related to the difference in the process parameters and the focusing condition. In particular, the focusing condition significantly influences the morphology of the molten tracks and the layers' surface roughness. This is in accordance with other authors who observed that a negative defocusing distance may lead to the formation of higher molten tracks with smaller contact angles, hence a higher surface roughness [45]. This can promote the formation of pores between the tracks and hinder powder deposition, causing the formation of small voids in the powder bed (Figure 5). These voids can in turn become pores with a shape and size similar to those observed in the tested samples. Although the porosity difference is low, it is possible that their presence can affect the mechanical properties of the material, in particular the fatigue resistance. The relative porosity, measured by Archimedes principle,

is <0.1 vol% for all the tested samples, which corresponds to a good density value for this kind of manufacturing technology. Eventually, the pore/void distribution was not significantly affected by the printing direction.

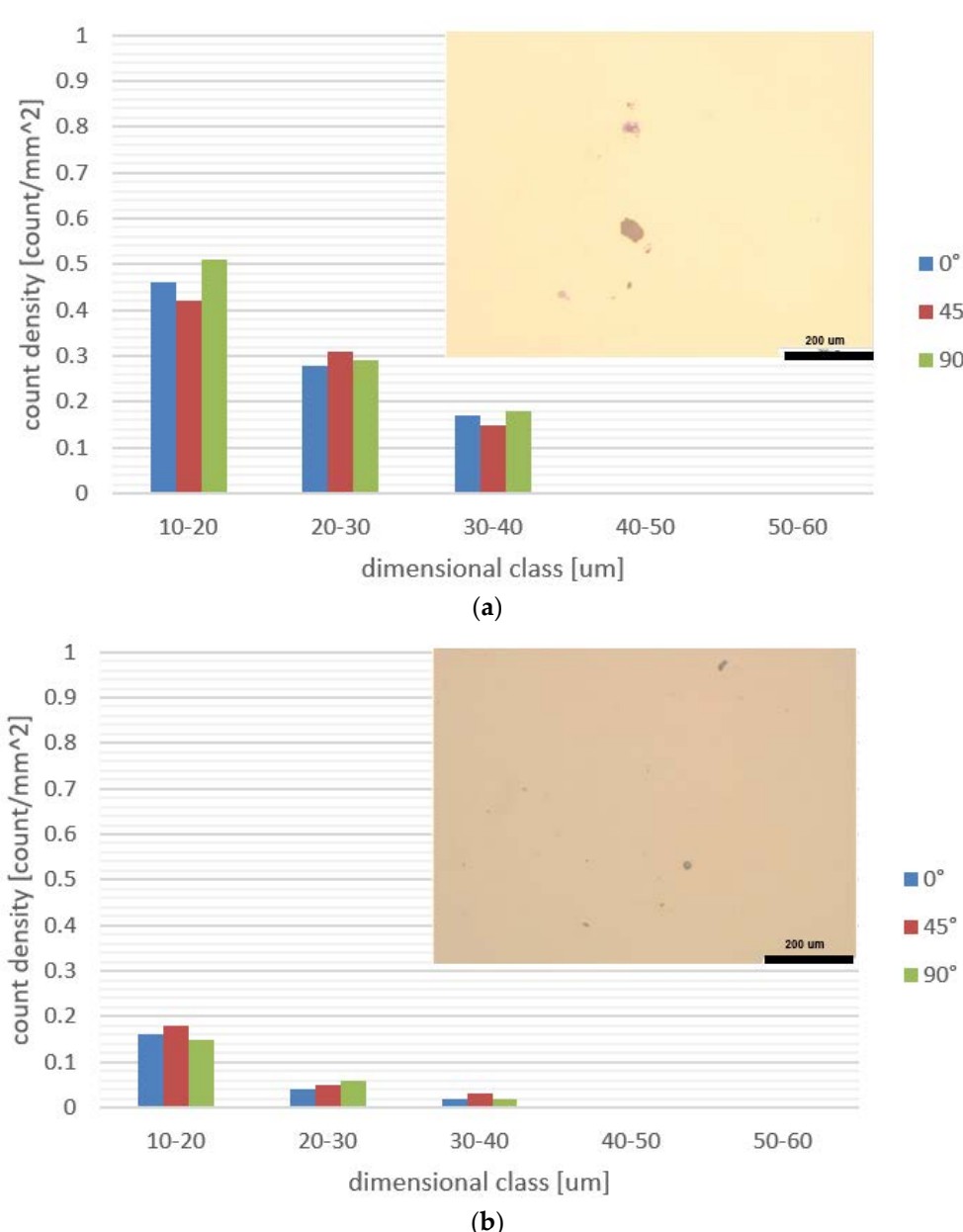

**Figure 4.** Pore/void distribution in samples produced with different build orientations on (**a**) the older and (**b**) the newer equipment. The images inside the graphs show the field of view with the highest porosity captured by a light microscope on representative samples.

In Figure 6, the bulk microstructure of the 3D printed materials is shown. The microstructure presented is only relevant in one printing direction (90°), close to the external surface and to the bulk. Indeed, no appreciable differences in microstructure between each printing direction were observed.

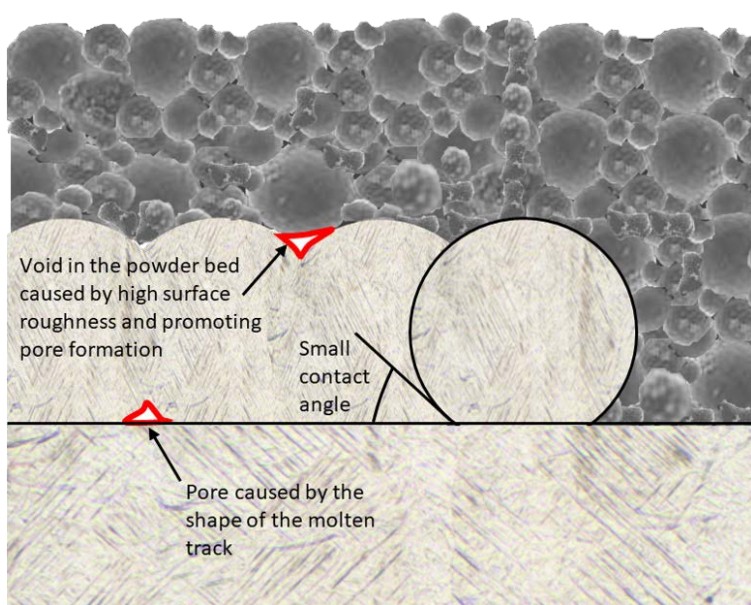

**Figure 5.** Pore formation due to the unfavourable shape of the molten tracks and voids in the powder bed induced by the layers surface roughness.

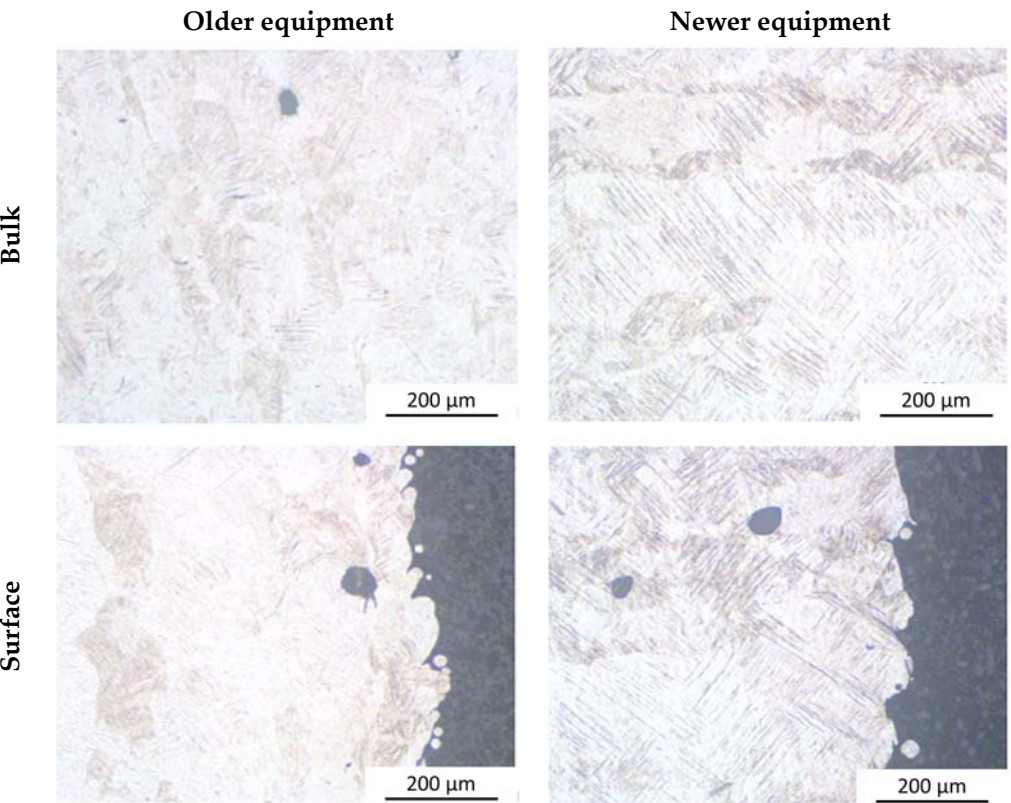

**Figure 6.** Microstructure of the analysed specimens (printing direction: 90°).

The microstructure of the analysed materials is composed entirely of Ti martensite, and no differences were found between the bulk and the surface for the same batch. It is possible to observe that the newer equipment has a Ti martensite produced by a more intense undercooling during solidification [10,21] that produces thicker martensitic laths compared to the martensite produced by older equipment. This is probably linked to the different energy densities of the two pieces of equipment used to produce the samples.

The microhardness profiles, acquired across the radius of the mechanical testing specimen, are shown in Figure 7.

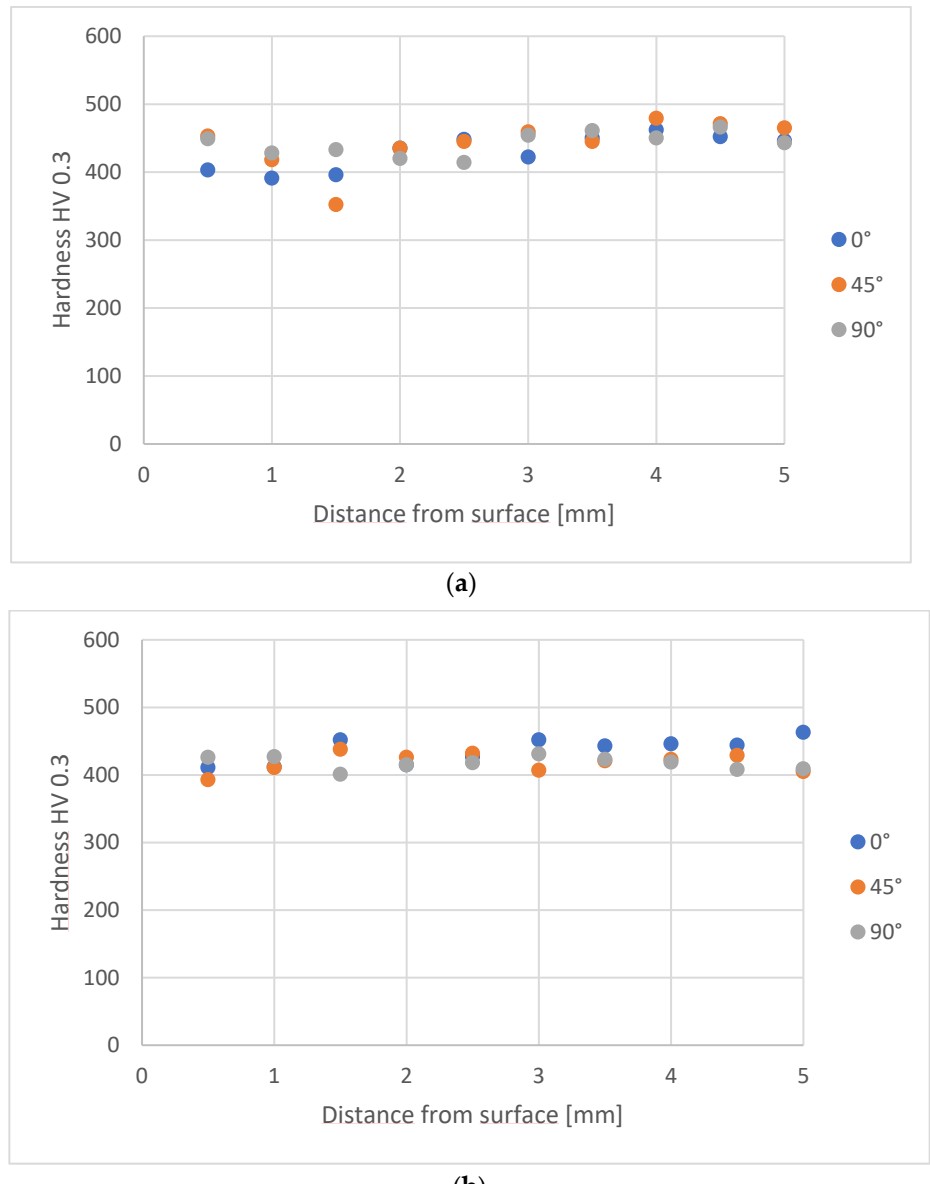

(**a**)

(**b**)

**Figure 7.** Microhardness profiles acquired across the diameter of a specimen used for mechanical testing: (**a**) older equipment and (**b**) newer equipment.

The microhardness profiles, for both the equipment and for each printing direction, are smooth and have an average hardness of about 430 HV. In this case, the slight difference in material microstructure has no effect on the hardness of the specimens. Moreover, there is no clear hardness variation versus distance from the sample surface.

### 3.2. Surface Morphology and Roughness

By analysing the surface morphology, many undercuts were noticed that were caused by the production process. The surface texture is strongly influenced by the layer-by-layer growth of material, as indicated in Figure 8. The undercuts are probably related to the overlay of the scan lines produced during the growth of the sample.

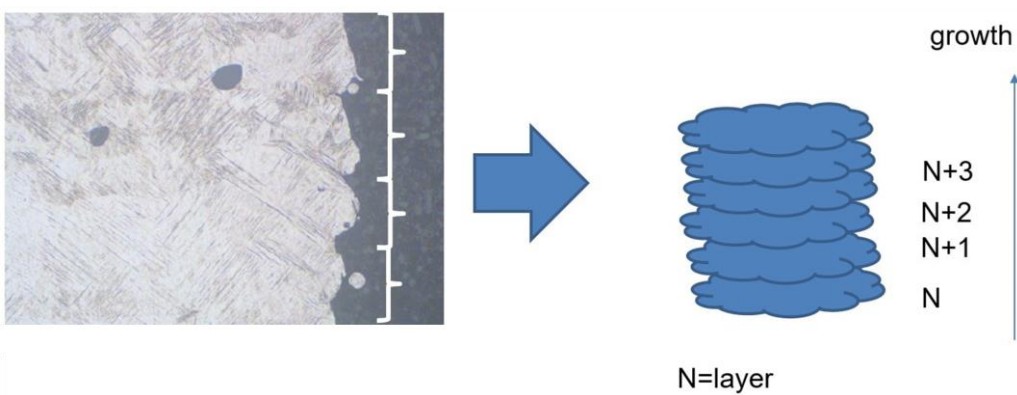

**Figure 8.** Scheme that indicates the correlation of the printing process to the surface texture.

This texture, induced by sample production, has a great effect on the surface roughness of the quasi-vertical samples (Figure 9).

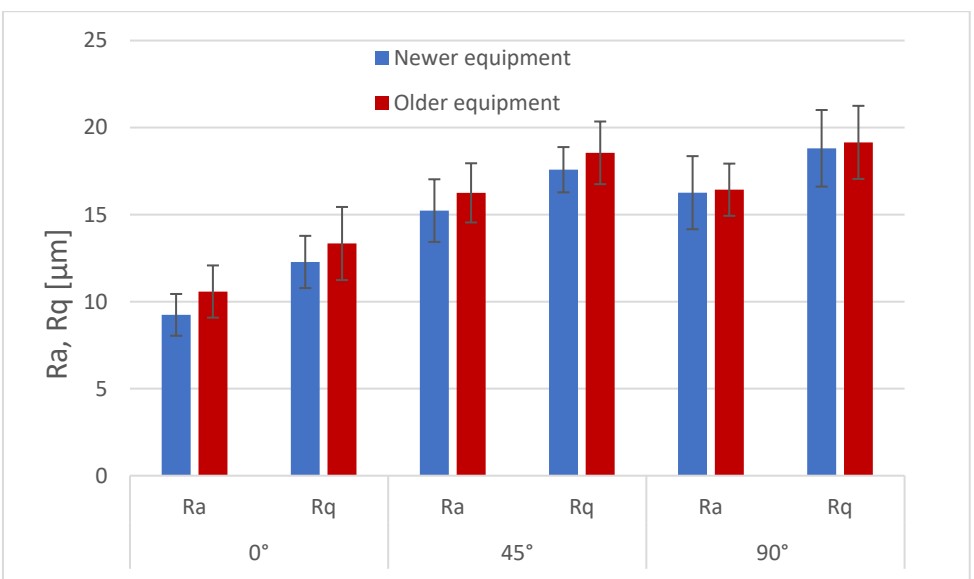

**Figure 9.** Roughness measurements (Ra and Rq) for the samples analysed in this work.

It is possible to observe that the Ra is strongly dependent on the printing direction. In particular, the horizontal samples show a smoother surface compared to the samples printed at 45° and in the vertical direction, while no significant differences were observed between the samples printed at 45° and 90°. The difference in surface roughness between the horizontal and 90°/45° directions is linked to the printing process. Indeed, the smoother surface of the 0° samples is related to the fact that the measured surface corresponds to the small laser scan lines. On the other hand, the rougher surface of the other printing directions is due to the presence of deep undercuts (Figure 8) or unmelted particles adhering to the surface. The results obtained are in agreement with previous experimental findings [18,20].

From the analysis of surface texture, it was possible to determine the severity of the surface by calculating the surface $K_t$ with Formula (1). The results of this calculation are presented in Figure 10.

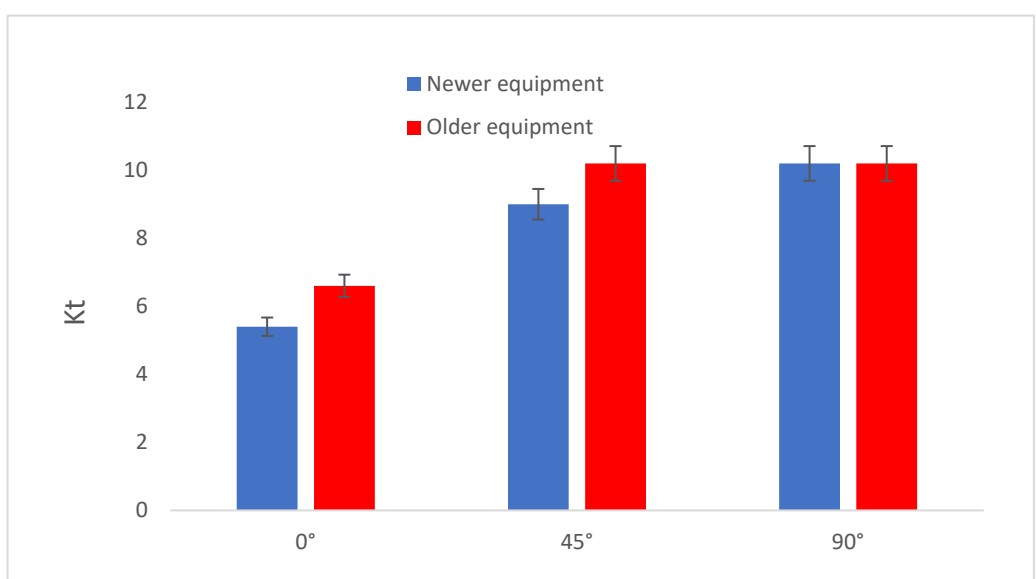

**Figure 10.** Comparison among $K_t$ values obtained from Formula (1) applied to the inspected specimens.

In this case, it is possible to observe that $K_t$ is lower for the specimens that were aligned with the building platform. This is in agreement with the Ra measurements presented in Figure 9, and it also explains the effect on surface texture. In addition, it is worth noting that the samples obtained with new equipment show a slightly lower $K_t$. The differences among the $K_t$ values can be explained by considering the different process conditions. In detail, the effect of the growth direction of the specimens is mainly related to the presence of the undercuts shown in Figure 8. They correspond to the overlay of the printed layer, and they are clearly visible in all the samples except the horizontal ones. The slightly different $K_t$ values obtained from the two machines are likely due to the different process parameters, especially the single layer thickness. This parameter is strongly dependent on the hatch distance and on the molten pool geometry, which in turn depend on the laser power, the scanning speed, and the laser spot diameter [26]. For a given shape of the molten pool, a disproportionate combination of layer thickness and hatch distance may result in the formation of a lack of fusion pores, according to the mechanism represented in Figure 11. At the same time, the layer thickness and the molten pool shape influence the formation of surface irregularities (Figure 11), possibly affecting the surface roughness and, in turn, the surface $K_t$. In particular, an increase in these parameters could increase the surface roughness and thus the surface $K_t$. In this regard, machine manufacturers tend to maximise the layer thickness to increase productivity, which was a severe problem for older machines working with small laser beams. It is likely that this affected the fatigue properties of the material.

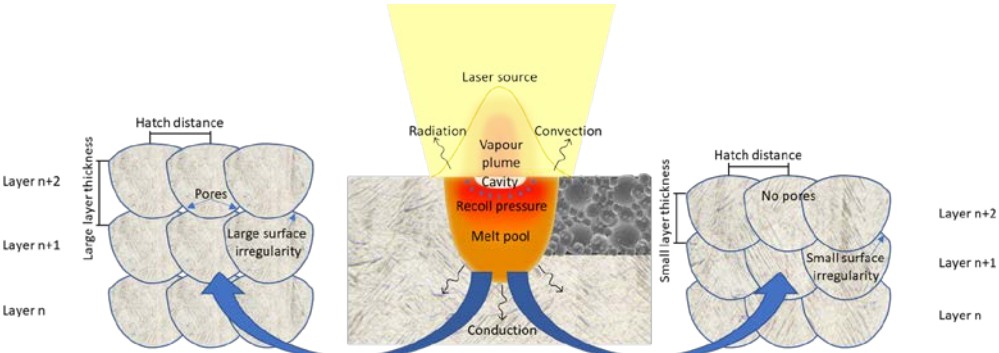

**Figure 11.** Relationship between the layer thickness, hatch distance, and molten pool shape and size and their effect on the surface quality.

### 3.3. Tensile and Fatigue Tests

The results of the tensile tests are presented in Figure 12.

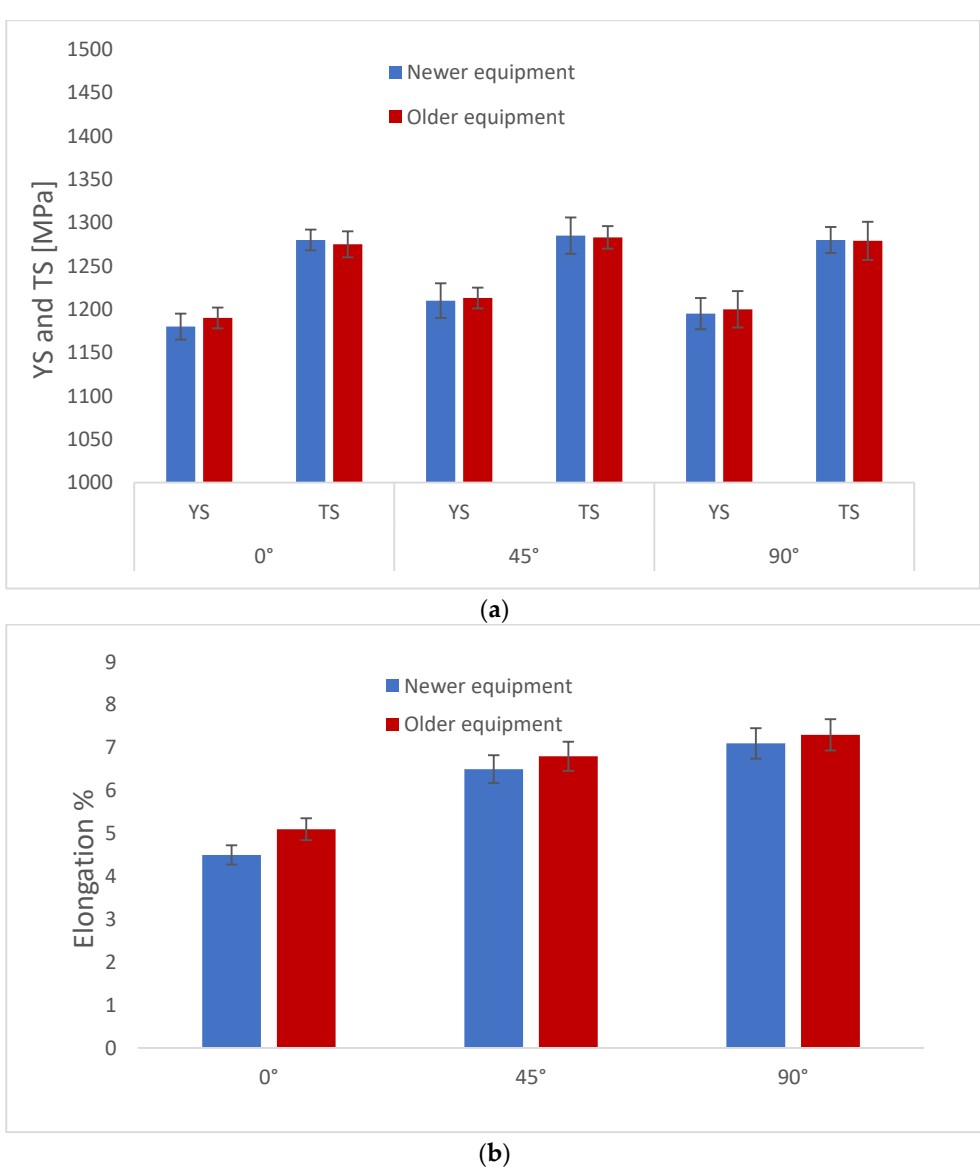

**Figure 12.** Tensile test results for the samples analysed in this work: Yield Strength (YS) and Tensile Strength (TS) (**a**) and Elongation (**b**).

From the analysis of the obtained results, Yield Strength (YS) and Tensile Strength (TS) do not significantly depend on the adopted production equipment, and neither depend on the printing direction. This result is probably related to the fact that the microstructure of the material (Ti martensite) is the same for all the samples tested, although a slight difference was detected between the martensite of the samples obtained by different printing techniques. The absence of differences between each printing direction is more related to the similarity of the microstructure of the material. This behaviour is also expected from the same hardness profiles observed in Figure 7. On the other hand, some differences were observed in terms of material ductility expressed through Elongation. Specifically, the horizontal samples show lower ductility compared to the vertical and 45° samples. This is probably a consequence of the macrostructure of the additively manufactured samples. As remarked by other researchers through EBSD analysis [21,32], the macrostructure of titanium alloys is generally composed of columnar macrograins, which are produced by epitaxial growth of metal during solidification. The orientation of macrograins affects

material ductility. In detail, material ductility is at its maximum when considering the vertical samples, whose macrograins are aligned with the applied load direction. This effect is not observed on TS and YS, in agreement with other works [18,24]. In this experience, no effects of the printing technique on ductility were detected.

The results obtained by uniaxial fatigue tests on the tested samples are shown in Figure 13.

In general, the horizontal samples (produced at 0°) exhibit better fatigue behaviour than the samples printed at 45° and 90°, before and after the fatigue limit elbow, for both production equipment. This is probably due to the lower surface roughness of horizontal samples with respect to those produced at 45° and 90°. In fact, the applied load during the uniaxial fatigue test is perpendicular to the undercuts observed in Figure 8 for the 45° and 90° specimens. No microstructural effects on fatigue life are expected since the quasi-static mechanical response is similar between the different specimens. The slight differences between the samples printed on older and newer equipment are more likely to be related to the internal void distribution or the slightly higher roughness found in the samples printed on older equipment. Using the Dixon–Mood approach (Figure 13c), a higher fatigue life was observed for the samples produced on the newer equipment. A possible reason for this better performance is the lower number of voids that characterise the samples produced by the newer equipment and the slightly lower surface roughness, as mentioned above. The fracture surfaces of some representative samples that failed at a stress level close to the fatigue limit are shown in Figure 14.

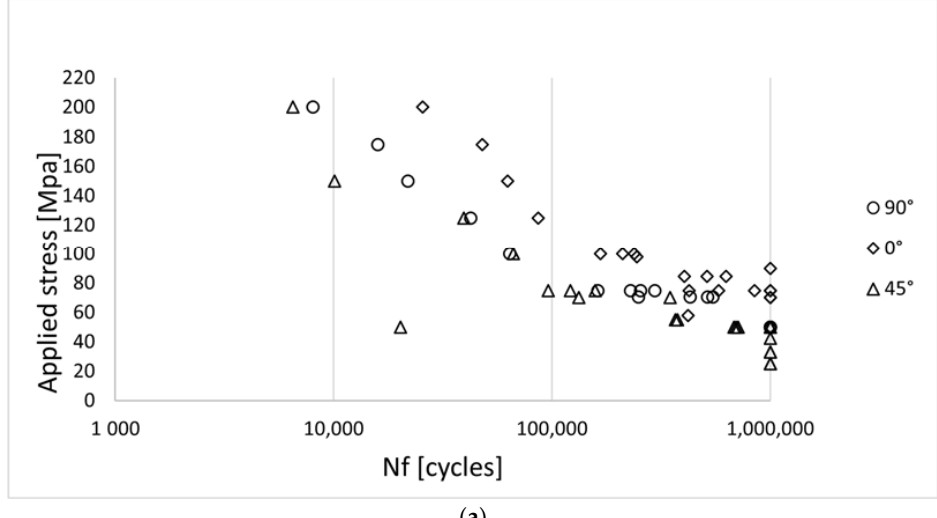

(**a**)

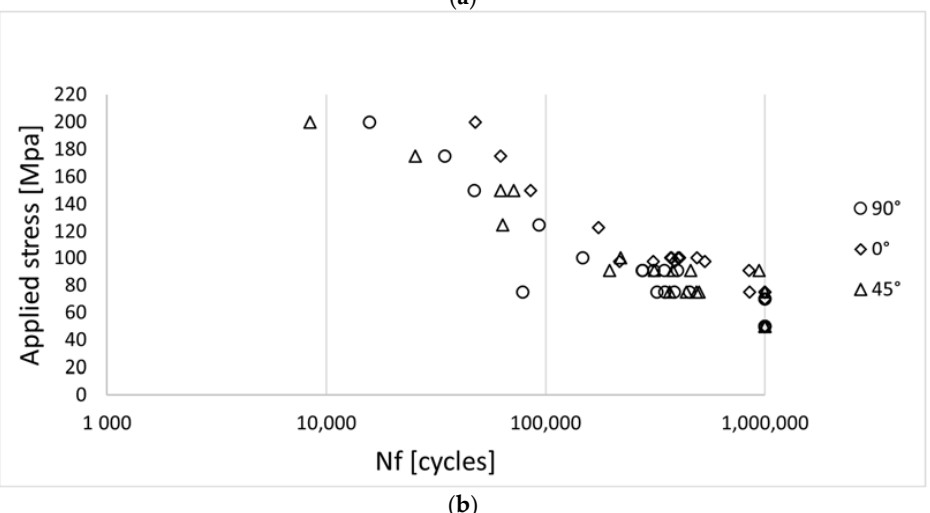

(**b**)

**Figure 13.** *Cont.*

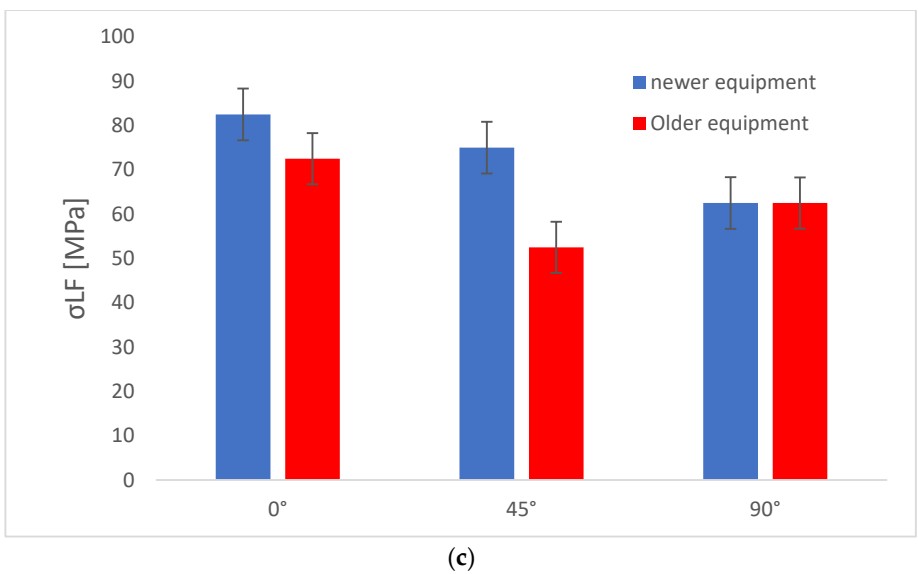

(**c**)

**Figure 13.** S-N curves of samples produced with older equipment (**a**), newer equipment (**b**), and fatigue limits comparison (**c**).

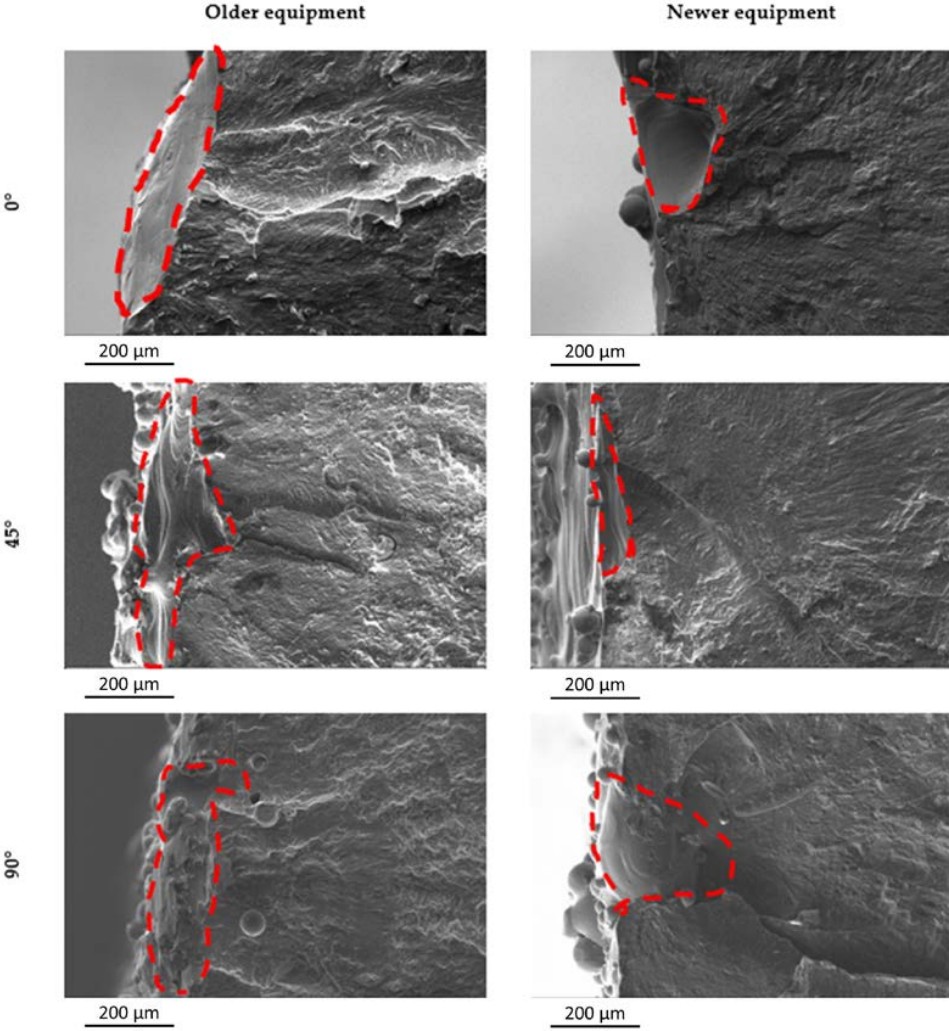

**Figure 14.** Representative fracture surfaces obtained on samples printed with the two different techniques and in the three different orientations. The fracture surfaces are related to samples tested at the same stress level (80 MPa).

The crack nucleation sites are all found near the external surface and on the opposite side with respect to the surface that was supported when producing the 0° and 45° samples. The surface defects are mainly related to morphologies similar to the undercuts observed and described in Figure 8. These discontinuities are related to an overlay of each molten layer produced during the 3D printing process. Larger defects (surface notches) were observed on the samples produced by the older equipment compared to those observed on the samples produced by the newer equipment. The reason is the different process parameters and dynamics, i.e., the different layer thickness and molten pool shape that influenced the formation of surface irregularities according to the mechanism described in Figure 11. At the older machine, a larger layer thickness combined with a lower laser power and a smaller and defocused laser beam were adopted, which may have determined slightly more unfavourable conditions, resulting in more detrimental surface defects.

The previously analysed defects also have an effect on the local fatigue limit calculated by means of the Murakami formula [44] close to the defect that caused the fatigue crack nucleation (Figure 15).

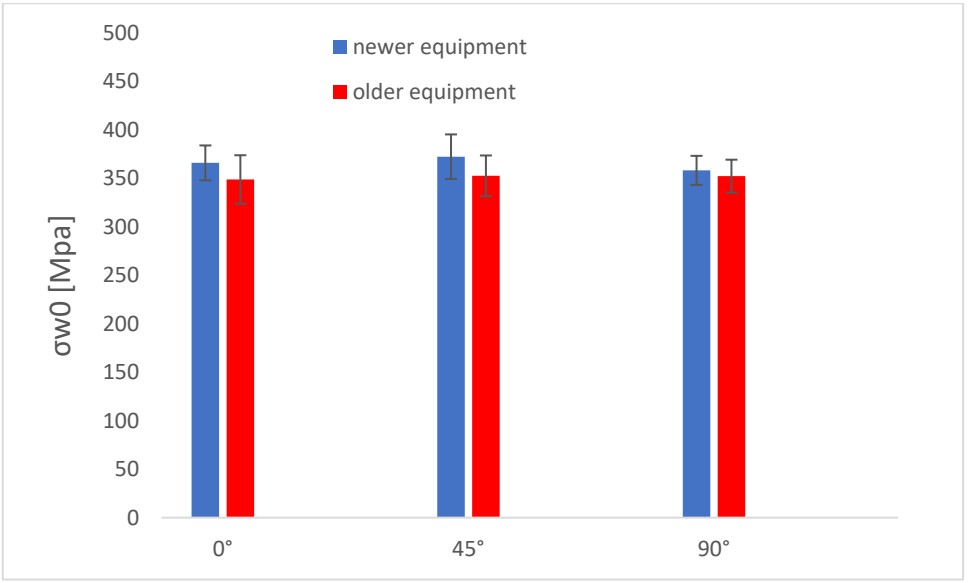

**Figure 15.** Local fatigue limit calculated by the Murakami approach for the tested samples as a function of printing direction.

It is possible to observe that the local fatigue limit is similar between all the tested samples and roughly seven times higher compared to the measured fatigue limit. This discrepancy between the local fatigue limit and the real fatigue limit can be correlated to the presence of residual stresses that have an effect on the fatigue behaviour of materials and are usually present on materials produced by the L-PBF process [4,20]. The local fatigue does not vary considerably when considering different printing directions; it is slightly smaller when considering the samples produced by the older equipment. This is probably related to the higher defects' extensions, which are mainly undercuts originated by the molten pool overlay found on broken specimens.

A rough evaluation of the local residual stresses—which were responsible for the fatigue crack nucleation (to be considered only in this area)—can be carried out by means of Walker's relation (3), as suggested by other authors [14]. For this purpose, the real $R_{real}$ should be acquired, which is correlated to the average applied stress corresponding in this work to the axial residual stresses of the tested samples. To calculate the $R_{real}$ parameter, both the $\sigma_{w0}$ and the $\sigma_{lf}$ were previously presented, while the n exponent was taken in agreement with Cutolo et al. and was equal to 0.385 [14]. The results obtained are shown in Table 3.

**Table 3.** $R_{real}$ values calculated from Walker's relation for the samples tested in this work.

| Printing Direction | Older | Newer |
| --- | --- | --- |
| 0° | 0.97 | 0.96 |
| 45° | 0.98 | 0.98 |
| 90° | 0.98 | 0.98 |

The calculated values for $R_{real}$ are close to one. In this experience, the fatigue tests were performed to R = −1. Thus, the axial residual stresses are close to the YS of the material. This justifies the low values of fatigue limit acquired by the fatigue tests, and it is in agreement with other tests on Ti gr. 23 in as-built conditions [14,19]. Eventually, the $R_{real}$ parameter was approximately constant independently of the printing direction and the production equipment used.

### 4. Conclusions

The present work aimed to determine the effect of 3D printing machine temporal evolution on the properties of the as-built Ti gr. 23 alloy that was produced by using three different growth directions.

The main results are summarised below:

- The microstructural characterisation showed that the samples printed with the older equipment had a larger amount of external and internal defects of coarser size compared to the samples printed with the newer equipment. The occurrence and number of internal defects are mainly correlated to the printing parameters;
- Surface texture analysis of samples printed on older equipment showed a rougher surface texture (about 10% higher). This is related to both the process parameters and the different optical systems of the laser;
- Tensile test results showed no difference in the YS, TS, or E% of the samples tested;
- Regarding the fatigue properties of the as-built specimens, in general, poor fatigue resistance was observed, especially for the specimens printed at 45° and 90° (10–20 MPa less compared to the horizontal specimens). This result is mainly due to the surface texture (especially the undercuts generated by the overlay redeposition) and the residual stresses rather than the presence of internal defects. However, the fatigue performance of the specimens produced on the newer equipment is slightly better than that of the specimens produced on the older equipment (10–20 MPa) for the horizontal and 45° specimens. This is mainly due to the higher quality of the surface, which is the typical crack initiation site for the specimens tested.

The observed results were a direct consequence of the improved equipment operation and the use of more powerful process parameters, which in turn were made possible by the novel equipment installed on the newer machine. Taken together, these results ultimately reflect the technological progress of the equipment.

**Author Contributions:** Conceptualisation, A.L. and E.V.; methodology, A.L.; validation, A.L., E.V., and G.T.; formal analysis, A.L., E.V., and M.M.; investigation, A.L. and M.M.; data curation, E.V. and G.T.; writing—original draft preparation, A.L. and E.V.; writing—review and editing, M.S., L.F., and G.T.; supervision, L.F. and M.S. All authors have read and agreed to the published version of the manuscript.

**Funding:** This study was carried out within the Interconnected Nord-Est Innovation Ecosystem (iNEST) and received funding from the European Union Next-Generation EU (PIANO NAZIONALE DI RIPRESA E RESILIENZA (PNRR)—MISSIONE 4 COMPONENTE 2, INVESTIMENTO 1.5—D.D. 1058 23/06/2022, ECS00000043). This manuscript reflects only the authors' views and opinions; neither the European Union nor the European Commission can be considered responsible for them.

**Data Availability Statement:** Restrictions apply to the availability of these data.

**Conflicts of Interest:** The authors declare no conflict of interest.

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
