# Peer review of "Study of the Effect of L-PBF Technique Temporal Evolution on Microstructure, Surface Texture, and Fatigue Performance of Ti gr. 23 Alloy"

_metals, doi:10.3390/met13071247_

Round 1
Reviewer 1 Report
In this paper, the effects of 3D printing machine temporal evolution on microstructure, surface texture, defect distribution and fatigue life are investigated. It is of great significance for improving the surface quality, the microstructure and the mechanical properties of the 3D printed parts. The article has certain innovation, but there are still some problems that need to be modified. There are many formatting problems in the article. The detailed comments are as follows:
1. The abstracts and conclusions in this paper need to be streamlined, concise and clear.
2. There is a problem with the scale in the paper such as Figure 1 c) and Figure 6. The dimensioning in Figure 3 a) is problematic.
3. The caption of Figure 4 needs to be explained in detail. In addition, the title of some figures in the text is not clear.
4. There should be a space between the unit and the number. In addition, there are many format problems in the text.
5. The definition of ρr in line 165 formula (1) needs to be explained.
6. The discussion on mechanical properties Yield Strength (YS) and Tensile Strength (TS) on line 270 is whether it should be combined with the microstructure characterization under this printing parameter, such as grain growth in Section 3.3.
Language needs to be polished, and there are some problems such as non-standard expression.
Author Response
Reviewer 1
Question: The abstracts and conclusions in this paper need to be streamlined, concise and clear.
Answer: the abstract was rewritten. The conclusions were revised.
Question: There is a problem with the scale in the paper such as Figure 1 c) and Figure 6. The dimensioning in Figure 3 a) is problematic.
Answer: Figures 1 and 6 were adjusted to enhance readability. However, the reasoning behind considering Figure 3 as problematic is unclear to the authors.
Question: The caption of Figure 4 needs to be explained in detail. In addition, the title of some figures in the text is not clear.
Answer: The caption of Figure 4 was revised to provide a clearer explanation of the graphs' significance.
Question: There should be a space between the unit and the number. In addition, there are many format problems in the text.
Answer: the entire text underwent a thorough review, and a space was added between each number and its corresponding units. Additionally, other formatting inconsistencies throughout the paper were addressed and rectified..
Question: The definition of ρr in line 165 formula (1) needs to be explained.
Answer: we added a reference that describes in detail all the procedure related to Kt calculation.
Question: The discussion on mechanical properties Yield Strength (YS) and Tensile Strength (TS) on line 270 is whether it should be combined with the microstructure characterization under this printing parameter, such as grain growth in Section 3.3.
Answer: The discussion was implemented in this paragraph.
Reviewer 2 Report
This paper investigates the microstructure, fatigue and tensile properties of Ti gr.23 alloy fabricated with additive manufacturing by two different models of Concept Laser M2 Causing machines. The data is comprehensive, but there are also some problems:
1. The logic of the introduction part is unclear, and it does not highlight the importance of this work. Thus, the introduction needs to be reorganized.
2. It can be seen from Table 2, the process parameters of older and newer equipment are different, then why did the authors choose this way? In this circumstance, what determines the difference in microstructures and mechanical properties? The process parameters or equipment?
3. It seems that the microstructures in Fig. 6 are different since the martensite is obvious in newer equipment, why the author said that “no appreciable differences on microstructure between…” in line 220?
4. 3.2 Part, what is the “surface texture”? Why do the horizontal samples demonstrate smoother surface compared with other two directions? The paper just displays results while lacks the mechanism analysis. Line 262, how do the layer thickness and molten pool shape influence the surface irregularities? The same as in line 278 “The orientation of macro grains affects material ductility”.
5. Line 281, what is the “grain size reinforcement effect”?
6. Line 211 “the porosity difference is low”, while line 298 “larger defects were observed on the samples……” Is this a contradiction?
7. The authors mention “residual stress”, while there is no introduction or any data of residual stress in the whole paper.
English needs to be improved.
Author Response
Reviewer 2
Question: The logic of the introduction part is unclear, and it does not highlight the importance of this work. Thus, the introduction needs to be reorganized.
Answer: the introduction was revised.
Question: It can be seen from Table 2, the process parameters of older and newer equipment are different, then why did the authors choose this way? In this circumstance, what determines the difference in microstructures and mechanical properties? The process parameters or equipment?
Answer: in the experimental paragraph we motivated the difference in process parameters ….” The main difference between the machines was the optical system, …. instead of equaling the energy input.”. This section was rewritten to provide a clearer understanding of the research aim and method.
The main aim of the paper was demonstrating that technological advances in LPBF influenced the properties of Ti gr23 components and thus the knowledge on 3D printed materials. This effect resulted from improving the equipment functioning itself and by including apparatus and systems enabling the use of more effective process parameters. We concluded (see conclusions) that in our opinion the main effect is due to the change in process parameters, however, it was possible only thanks to the development of the machines and the use of novel apparatus enabling them. This argument was better explained in the conclusions to effectively convey the results of our research.
Question: It seems that the microstructures in Fig. 6 are different since the martensite is obvious in newer equipment, why the author said that “no appreciable differences on microstructure between…” in line 220?
Answer: The specified sentence has been modified. The microstructure has been described and the differences between the 2 samples have been better highlighted.
Question: 3.2 Part, what is the “surface texture”? Why do the horizontal samples demonstrate smoother surface compared with other two directions? The paper just displays results while lacks the mechanism analysis. Line 262, how do the layer thickness and molten pool shape influence the surface irregularities? The same as in line 278 “The orientation of macro grains affects material ductility”.
Answer: In this part we have described the mechanisms through the Kt evaluation, which contains all the information about the surface texture (Formula 1). We have added a sentence to clarify this point. We have also added a sentence explain why the horizontal samples are smoother and to clarify how the variation of process parameters can affect the final result in terms of Kt and therefore surface roughness. However a brief discussion on surface texture was added in the part of Ra and Rq description.
Question: Line 281, what is the “grain size reinforcement effect”?
Answer: The sentence has been deleted because, as the referee has demonstrated, it adds no useful information to the text.
Question: Line 211 “the porosity difference is low”, while line 298 “larger defects were observed on the samples……” Is this a contradiction?
Answer: The internal porosity of the material is low and this observation applies to the majority of the material. The larger defects are associated with surface undercuts (we have clarified this in the sentence). There is no contradiction in the 2 sentences.
Question: The authors mention “residual stress”, while there is no introduction or any data of residual stress in the whole paper.
Answer: We partially agree with the reviewer. Indeed, we did not intend to analyse the residual stress of the material, but we justified the discrepancies between the fatigue data and the local fatigue limit by means of a relationship that qualitatively gives a rough indication of the amount of residual stress. In this case, we emphasized that the as-printed material is likely to have residual stresses very close to the yield strength of the material in the region of crack nucleation. We have clarified this point in the text and in Materials and Methods.
Reviewer 3 Report
The work has more applied than scientific value. Currently, the article requires major revision and is weak in its current form for publication in Q1 journal
some comments on the work below
Abstract should not be divided into 2 parts. In the abstract, it is necessary to indicate the scientific novelty and scientific significance of the work, add the numerical values of the studies
Contacts of the authors are not presented according to the format of the journal
The introduction is written too generally. There are abstract proposals with reference to more than 10 works: «Both of them are processed by several additive manufacturing 48 techniques such as L-PBF (Laser-Powder Bed Fusion) [1,2,11–20,3,21–30,4,31,32,5–10] and 49 EBM (Electron Beam Melting) [16,22,33–35].»
It is necessary to conduct a more detailed analysis of publications, point out the specific results of previous studies
The possibilities of heat treatment should be mentioned, for example, Study on Changes in Hardness and Wear Resistance of 3D Printed Ti6Al4V with Heat Treatment Temperature. Tribology in Industry, 44(1), 129 and Structure and Properties of Ti–50.2 Ni Alloy Processed by Laser Powder Bed Fusion and Subjected to a Combination of Thermal Cycling and Heat Treatments. Shape Memory and Superelasticity, 8(1), 16-32.
Purpose and scientific novelty should be spelled out more clearly
Figures and tables are not made according to the format of the magazine
Many of the drawings are of very poor quality.
Figure 9, 10, 11, 12, 15 pasted from excel
The presentation format of Figures 9-15 should be rethought and presented in one figure.
The Tensile and fatigue tests section requires more discussion, it is necessary to present several initial diagrams with the designation of the analyzed values
The designations "Newer equipment" "Older equipment" in the opinion of the reviewer are not very suitable. In the methodology, it is necessary to indicate the fundamental differences between the equipment
The discussion in the article requires a more detailed analysis with an explanation of the physical nature of the observed patterns.
The conclusions are written too generally and do not fully reflect the results achieved. Numeric values need to be added
Author Response
Reviewer 3
Question: Abstract should not be divided into 2 parts. In the abstract, it is necessary to indicate the scientific novelty and scientific significance of the work, add the numerical values of the studies
Answer: the abstract was rewritten.
Question: Contacts of the authors are not presented according to the format of the journal
Answer: @mails added;
Question: The introduction is written too generally. There are abstract proposals with reference to more than 10 works: «Both of them are processed by several additive manufacturing 48 techniques such as L-PBF (Laser-Powder Bed Fusion) [1,2,11–20,3,21–30,4,31,32,5–10] and 49 EBM (Electron Beam Melting) [16,22,33–35].»
It is necessary to conduct a more detailed analysis of publications, point out the specific results of previous studies
The possibilities of heat treatment should be mentioned, for example, Study on Changes in Hardness and Wear Resistance of 3D Printed Ti6Al4V with Heat Treatment Temperature. Tribology in Industry, 44(1), 129 and Structure and Properties of Ti–50.2 Ni Alloy Processed by Laser Powder Bed Fusion and Subjected to a Combination of Thermal Cycling and Heat Treatments. Shape Memory and Superelasticity, 8(1), 16-32.
Purpose and scientific novelty should be spelled out more clearly
Answer: The introduction has been revised. A citation of heat treatment on mechanical properties of Ti alloy has been added in the Introduction.
Question: Figures and tables are not made according to the format of the magazine
Answer: Figures and tables are now in the format required by the journal. The problems were related to the transfer of this information from a different format to that of the Journal.
Question: Many of the drawings are of very poor quality.
Answer: We have added schemes of several phenomena to describe some phenomena related to the printing process. The term very poor quality is related to image resolution or what else?
Question: Figure 9, 10, 11, 12, 15 pasted from excel
Answer: If we have understood well, the format of the figures is not the same. Now we have standardised the format.
Question: The presentation format of Figures 9-15 should be rethought and presented in one figure.
Answer: If we have understood correctly, we have to summarise the figures 9 to 15 in one figure. We do not agree with the reviewer just because the figures show different parameters and are part of different paragraphs presented in the article.
Question: The Tensile and fatigue tests section requires more discussion, it is necessary to present several initial diagrams with the designation of the analyzed values
Answer: The discussion was implemented in the tensile and fatigue tests part.
Question: The designations "Newer equipment" "Older equipment" in the opinion of the reviewer are not very suitable. In the methodology, it is necessary to indicate the fundamental differences between the equipment
Answer: The differences between new and old equipment are well evidenced in materials and method paragraph. “The main difference between the machines was the optical system, which in the older model was composed of the collimator, the beam expander, the scanning head and the F-theta lens, while in the newer model it also included a dynamic focusing unit for adjusting the laser spot size. Consequently, radically different process parameters (see Table 2) were recommended by the machine manufacturer when processing the considered Titanium alloy in the two machines types. In particular, a small laser beam in a defocused condition was recommended at the older machine while a much larger but focused laser beam was recommended at the newer machine endowed with the focusing unit. Process parameters differ from each other both taken separately and when combined in synthetic indicators describing the energy density. The most similar physical quantity in the two cases was the Linear Energy Density (LED) [29], which was 17% higher on the bulk areas and 27% lower on the contours with the new machine. The Surface Energy Density (SED) [35] and the Volumetric Energy Density (VED) [30] were instead remarkably lower. However, the energy density is not a good indicator for comparing different process parameters [40], since it neglects the main effects and interactions among the considered factors. In addition, it also miss to capture the influence of defocusing, which is a main feature of the tested equipment. For that reasons, it was decided to compare the performance of the machines in their respective optimum conditions, instead of equaling the energy input…..”. This section was rewritten to provide a further clearer understanding of the differences among the machines.
Question: The discussion in the article requires a more detailed analysis with an explanation of the physical nature of the observed patterns.
Answer: we discussed more in detail the obtained results.
Question: The conclusions are written too generally and do not fully reflect the results achieved. Numeric values need to be added
Answer: The conclusions were revised and we added some numerical values to better highlight the differences among the tested machines,, and we emphasized the effect of the technological progress of the equipment to effectively convey the results of our research.
Round 2
Reviewer 3 Report
The authors of the article significantly revised the article. The responses to the comments satisfied the reviewer.
Nevertheless, I would like to emphasize that this research is more of an applied nature, the scientific component is small.
There are also comments related to the design of the article and, first of all, with drawings. I hope the staff of the magazine will make a good layout of the article